# Seasonality and Vertical Structure of Microbial Communities in Alpine Wetlands

**DOI:** 10.3390/microorganisms13050962

**Published:** 2025-04-23

**Authors:** Huiyuan Wang, Yue Li, Xiaoqin Yang, Bin Niu, Hongzhe Jiao, Ya Yang, Guoqiang Huang, Weiguo Hou, Gengxin Zhang

**Affiliations:** 1Institute of Earth Sciences, China University of Geosciences, Beijing 100083, China; wanghy@itpcas.ac.cn (H.W.); 15755401397@163.com (Y.L.); 17733479782@163.com (Y.Y.); 18831620631@163.com (G.H.); 2Institute of Tibetan Plateau Research, Chinese Academy of Sciences, Beijing 100101, China; yangxiaoqin@itpcas.ac.cn (X.Y.); niubin@itpcas.ac.cn (B.N.);; 3University of Chinese Academy of Sciences, Beijing 100049, China; 4State Key Laboratory of Biogeosciences and Environmental Geology, China University of Geosciences, Beijing 100083, China

**Keywords:** alpine wetlands, depth, seasonal variation, soil microbes

## Abstract

The soil microbial community plays a crucial role in the elemental cycling and energy flow within wetland ecosystems. The temporal dynamics and spatial distribution of soil microbial communities are central topics in ecology. While numerous studies have focused on wetland microbial community structures at low altitudes, microbial diversity across seasons and depths and their environmental determinants remain poorly understudied. To test the seasonal variation in microbial communities with contrasting seasonal fluxes of greenhouse gases, a total of 36 soil samples were collected from different depths in the Namco wetland on the Tibetan Plateau across four seasons. We found significant seasonal variation in bacterial community composition, most pronounced in the Winter, but not in archaea. In particular, Proteobacteria decreased by 11.5% in Winter compared with other seasons (*p* < 0.05). The bacterial alpha diversity showed hump-shaped seasonal patterns with lower diversity in Winter, whereas archaea showed no significant patterns across depths. A PERMANOVA further revealed significant differences in the bacterial community structure between Winter and the other three seasons (*p* < 0.05). In addition, bacterial and archaeal community structures differed between surface (0–5 cm) and deeper (5–30 cm) soils (*p* < 0.01). Redundancy analysis showed that soil total nitrogen, soil total phosphorus, and total soil organic carbon significantly influenced bacteria and archaea (*p* < 0.05). Furthermore, soil moisture content and temperature strongly affected the bacterial community structure (*p* < 0.001). Our findings highlighted the seasonal variation in the microbial community and the profound influence of soil moisture and temperature on microbial structure in alpine wetlands on the Tibetan Plateau.

## 1. Introduction

Soil microorganisms play a relevant role as decomposers in ecosystems, regulating nutrient circulation and energy flow [1]. Their community composition and diversity exhibit strong geographic dependency: in temperate peatlands, seasonal temperature fluctuations dominate microbial dynamics [2], whereas in arid wetlands, water availability primarily regulates microbial activity across soil depths [3]. Recent studies further highlight contrasting drivers of microbial vertical distribution—for example, oxygen diffusion controls stratification in waterlogged subtropical marshes [4], while root exudates shape depth-dependent patterns in mangrove forests [5]. However, how seasonal precipitation interacts with soil depth to drive microbial succession remains poorly understood in seasonally flooded wetlands, particularly regarding whether carbon substrate availability or redox potential is the ultimate constraint. This knowledge gap limits the predictive understanding of wetland biogeochemical responses to climate extremes.

Wetland microbial communities are dynamic engines of biogeochemical cycles, whose spatiotemporal patterns reflect complex adaptations to environmental gradients. Seasonal and vertical diversity shifts, observed across global wetland ecosystems, are increasingly linked to taxon-specific functional traits that mediate carbon and nutrient fluxes. Some studies have shown that the soil microbial community exhibited significant seasonal variation, with the highest diversity in Spring and the lowest in Winter in the Zhalong wetland [4]. Furthermore, some research found that the seasonal variation in microbial communities was mainly affected by regional temperature and available substrates [6]. In Arctic peatlands, *Methanosarcina* dominance during Summer thaw drives methane emissions through temperature-sensitive acetate fermentation pathways, while Winter freezing enriches cold-adapted Methanococcales reliant upon hydrogenotrophic methanogenesis [7]. Such metabolic specialization underscores how microbial energy allocation strategies—balancing substrate acquisition and stress tolerance—shape community assembly under fluctuating conditions. These principles resonate in subtropical wetlands yet manifest through distinct biogeochemical feedback mechanisms. Similar redox-driven stratification occurs in China’s Yellow River Delta wetlands, where methanogens progressively outcompete sulfate reducers with soil depth, resembling Siberian peatland dynamics [8]. Alpine wetlands, however, present an understudied paradox: despite prolonged hypothermia, methane emissions persist through poorly constrained mechanisms. Concurrently, the low-temperature suppression of methane-oxidizing *Methylocystis* may decouple methanogen diversity from CH_4_ flux magnitudes [9]. Such adaptations highlight context-dependent trade-offs between community diversity and functional resilience—a phenomenon observed in European fens where Winter Actinobacteria dominance is enabled by cold-adapted lignocellulose systems [10]. Critical gaps persist in resolving how alpine wetland microbes allocate energy between stress survival and biogeochemical functions. Specifically, it remains unclear whether seasonal diversity declines reflect the functional redundancy loss or adaptive enrichment of keystone taxa capable of multifunctional metabolisms under poly-extreme conditions.

The Qinghai–Tibet Plateau contains 1.79 × 10^5^ km^2^ of wetlands, accounting for 33.39% of China’s total wetland area [11]. Due to the low temperature and thin oxygen in high-altitude areas [12], alpine wetlands are vulnerable and sensitive to climate change and human activities [13]. A recent study showed that large areas of alpine wetlands are experiencing a rapid mineralization of soil organic carbon and a rapid loss of soil nutrients [14]. In this study, soil samples from four seasons at 0–5 cm, 5–10 cm, and 10–30 cm were collected in the Namco wetland. (1) We hypothesize that soil microbial community composition and functional structure exhibit distinct seasonal variations across different soil horizons, with surface layers demonstrating greater temporal turnover compared to deeper soil strata. This seasonal stratification is predicted to be the most pronounced in ecosystems experiencing seasonal freezing–thawing cycles and pronounced hydrological fluctuations. (2) The vertical distribution patterns of microbial taxa are primarily governed by the interplay between soil physicochemical gradients (including organic carbon availability, pH, and moisture content) and microbial life history strategies.

## 2. Materials and Methods

### 2.1. General Situation of Research Area

The Namco wetland (29°57′–30°33′ N, 89°49′–90°17′ E) is located between Damxung County of Lhasa City and Bango County of Nagqu Prefecture, with an area of 1961 km^2^ and an elevation of 4720 m above sea level (Figure 1a). The sampling site is covered with an alpine swampy meadow of *Carex tibetikobresia* (Cyperaceae), *Kobresia hohxilensis* (Cyperaceae), and *Kobresia humilis* (Cyperaceae) with an average height of 15–20 cm and a total coverage of about 60–80% [15]. The region belongs to the typical plateau continental climate. The annual mean temperature is 0 °C, with the coldest monthly temperature occurring in January (mean temperature −18.2 °C) and the warmest monthly temperature in July (mean temperature 14.7 °C, Appendix A). The annual precipitation is primarily concentrated in the period from May to October, with distinct rainy and dry seasons [16]. The temperature in 2017 used in this study was recorded at the NamCo Multi-layer Comprehensive Observation and Research Station (Figure 1b).

### 2.2. Sample Collection

Soil samples were collected from 3 layers (0–5 cm, 5–10 cm, 10–30 cm) from the Namco wetland on 23 January, 23 June, 25 August, and 6 October 2017 and homogenized with a sterilized spoon. After discarding the visible stones and the plant roots, 500 g of soil was sieved through a 2 mm mesh and then stored at 4 °C for physiochemical analyses or at −80 °C for microbial community analyses.

### 2.3. Determination Methods

#### 2.3.1. Determination of Soil Physical and Chemical Properties

Five grams of soil was added to distilled water (1:2.5, soil–water ratio), after which the mixture was stirred for 1 min and stood still for 30 min, and the pH was determined with a pH meter (Sartorius PB-10, Sartorius AG, Göttingen, Germany). A total of 5 g of natural air-dried soil was sieved through a 1 mm sieve and incubated in a 105 °C oven to constant weight; the lost weight was measured, and the water content was calculated. The total organic carbon (TOC) and total nitrogen (TN) were measured with a CHNS/O Analyzer (PerkinElmer 2400 Series II, PerkinElmer Inc., Waltham, MA, USA) after 1N-HCl acidification and 50 °C oven-drying. Total phosphorus (TP) was measured by the molybdenum antimony resistance method [17]. Dissolved organic carbon (DOC) and total dissolved nitrogen (TDN) were measured by dissolving water-soluble substances in water by mixing soil in water with a weight ratio of 1:5 [17]. After conducting soil–water mixing and acidification with a drop of phosphoric acid, the soil was shaken for 120 r/min at room temperature for one hour [18]. The contents of DOC and TDN in the supernatant were determined by an N/C analyzer (Multi N/C 2100S, Analytik Jena AG, Jena, Germany).

#### 2.3.2. High-Throughput Sequencing of 16S rRNA Gene in Soil Microbial Community

The soil metagenomic DNA was extracted using a FastDNA^®^ SPIN kit for Soil (MP Biomedicals, Shrewsbury, UK). One microliter of DNA was amplified by a PCR using a 96-Well veriti Thermal Cycler (Thermo Fisher Scientific, USA) with forward primer 515F (5′-GTGCCAGCMGCCGCGGTAA-3′) and the reverse primer 806R (5′-GGACTACHVGGGTWTAAT-3′) to amplify the 16S rRNA gene V4 region [19]. The PCR products were loaded into 1% agarose gel and run for 30 min at 120 V voltage. The amplified fragments were recovered and purified using SV Gel and PCR Clean-Up System. The purified PCR products were dissolved in 20 μL purified water and quantified by using a Picogreen reagent (Quant-iT tm Picogreen DNA Reagent and Kits, Invitrogen, Carlsbad, CA, USA) and sequenced by the sequencing Chip (318 tm Chip V2, cat. No. 4484354) placed at the designated position of the IonTorrent PGM Sequencer (Thermo Fisher Scientific Inc., Waltham, MA, USA) for high-throughput sequencing according to the manufacturer’s protocol.

Purified amplicons were combined in equimolar ratios and subjected to paired-end sequencing using either the Illumina MiSeq PE300 platform or the NovaSeq PE250 platform (Illumina, San Diego, CA, USA). This process was carried out in accordance with the standard protocols established by Majorbio Bio-Pharm Technology Co. Ltd. (Shanghai, China), which are specifically optimized for microbial community profiling [20]. The raw sequence data, obtained in FASTQ format from 36 soil samples, underwent rigorous quality control and adapter trimming using fastp (version 0.20.0) [21]. Subsequently, the sequences were merged using FLASH, following previously validated bioinformatics pipelines [22]. Operational Taxonomic Units (OTUs) were generated with a 97% sequence identity threshold through de novo clustering, utilizing the UPARSE algorithm [23]. Chimeric sequences were identified and filtered out using USEARCH (version 7.0), which employs the UCHIME algorithm [24]. To ensure accurate taxonomic classification, representative OTUs were classified using the RDP Classifier against the Silva SSU database [25,26], with a confidence threshold of 70% applied to maintain annotation precision. Furthermore, sequence alignment and phylogenetic placement were validated through comparative analysis against reference datasets curated within the Silva repository [27]. This comprehensive approach ensures the reliability and accuracy of the microbial community profiling results.

### 2.4. Data Analysis

A one-way ANOVA was used to test the significance of seasonal and vertical variations in soil physical and chemical properties with SPSS 18.0. The Chao1 index was used to represent alpha diversity. Multivariate variance analysis (PERMANOVA) based on the Bray–Curtis distance matrix was used to test seasonal and vertical variation in microbial community dissimilarity. Redundancy analysis (RDA) was used to determine the impact of climate variables and soil physiochemical properties on the microbial community structure. A histogram of microbial relative abundance was drawn by OriginPro 2018. Most statistical analyses and plots were generated using R statistical software V3.6.3 with the packages vegan and ggplot2.

## 3. Results

### 3.1. Soil Physicochemical Properties in Alpine Wetlands

Soil physicochemical properties varied along depth and different seasons. For example, the average contents of TDN, TOC, and SMC in 0–5 cm depth soil in Spring, Summer, and Fall were 4.78 mg/kg, 0.13 mg/kg, and 0.018 mg/kg, respectively, which were higher than the respective contents of 4.1 mg/kg, 0.09 mg/kg, and 0.013 mg/kg in Winter. On the contrary, pH was the highest in the Winter soil depth. At different depths in the same season, soil physicochemical properties showed a general decreasing trend (Table 1), especially for the depth of 10–30 cm. For example, in June, the average content of TN, TP, and DOC at 0–10 cm was 10.21 mg/kg, 1.73 mg/kg, and 26.99 mg/kg, while the contents of these properties at 10–30 cm were only 8.03 mg/kg, 1.34 mg/kg, and 18.01 mg/kg, respectively.

### 3.2. Soil Microbial Community Composition in Alpine Wetlands Under Different Soil Depths and Seasons

After quality filtering and OTU clustering at 97% sequence identity, a total of 1,526,359 high-quality sequences were obtained from 36 soil samples. To reduce the bias caused by sequencing depth variation, all samples were rarefied to 21,359 reads per sample, resulting in 768,924 sequences used for downstream analyses. Based on taxonomic classification, 17,277 OTUs were identified across all samples. These included bacterial OTUs assigned to 25 phyla, 50 classes, 66 orders, 128 families, and 214 genera and archaeal OTUs classified into 2 phyla, 4 classes, 7 orders, 9 families, and 8 genera. Overall, bacteria accounted for 97.4%, and archaea accounted for 1.5%. Within bacteria, the dominant phyla included Proteobacteria (relative abundance 50.2%), Acidobacteria (12.6%), Actinomycetes (10.4%), Verrucomicrobia (5.8%), Bacteroidota (3.5%), Bacillota (3.0%), and Planctomycetes (8.5%) (Figure 2a). Proteobacteria mainly comprised Alphaproteobacteria (20.8%), Betaproteobacteria (17.8%), Gammaproteobacteria (3.0%), and Deltaproteobacteria (4.5%). From the perspective of seasonal variation, the relative abundances of the main phyla in Spring, Summer, and Fall did not change significantly. Still, the relative abundances of the dominant bacteria differed in Winter; the dominant Proteobacteria phylum especially decreased significantly (*p* < 0.05), from 41.5% to 11.5%, among which Alphaproteobacteria decreased by 5.1% in Winter compared with the other three seasons. The three dominant classes of archaea were Thermoprotei (92.21%), Methanomicrobia (4.85%), and Methanobacteria (2.94%) (Figure 2b). The relative abundance of Thermoprotei in Winter was 77.7%, with an increase of 5.7%, 10.0%, and 8.1% in comparison with Spring, Summer, and Fall, respectively. In terms of soil depth, there was no significant change in bacterial composition at 0–5 cm, 5–10 cm, and 10–30 cm, and the relative abundance of Thermoprotei at the 0–5 cm depth was 0.7% and 0.2% higher than that of 5–10 cm and 10–30 cm. At the same time, both Methanomicrobia and Methanobacteria decreased by around 0.33%. Overall, the relative abundance of soil microbial composition changed more significantly among the seasons than the soil depths. In contrast, the relative abundance of bacteria changed more considerably with the seasons than archaea.

### 3.3. Soil Microbial Alpha Diversity in Alpine Wetlands Under Different Soil Depths and Seasons

The Chao1 index is an indicator for estimating the total number of species of soil samples, and the higher the value, the higher the diversity of species in the sample. Overall, the alpha diversity of bacteria was higher than that of archaea, and the diversity of bacteria and archaea at different depths did not change significantly among the seasons. However, there were significant differences in the bacterial diversity of the depths of 0–5 cm and 5–10 cm in Winter compared with the other three seasons (*p* < 0.01), while the diversity of 10–30 cm showed significant differences between Spring and the other three seasons (*p* < 0.05) (Figure 3a). There were no significant differences in archaea in different seasons (*p* < 0.05) (Figure 3b).

### 3.4. Soil Microbial Beta Diversity in Alpine Wetlands Under Different Soil Depths and Seasons

A PERMANOVA revealed the differences in bacterial and archaeal community structures with respect to season and soil depth. The bacterial community exhibited significant temporal decay rates (Figure 4a). In contrast, archaea had no obvious seasonal patterns (Figure 4b). Furthermore, the dissimilarity of bacterial and archaeal communities exhibited increasing patterns with increasing depth in Spring (*p* < 0.001, Figure 4c,d). The microbial community structure (including bacteria and archaea) between the surface soil (0–5 cm) and deep soil (5–30 cm) differed significantly (*p* < 0.01). As for the variation in the seasons, the bacterial community structure in the Winter differed significantly from other seasons (*p* < 0.01), while the archaeal community structure did not change significantly with seasons (Table 2).

### 3.5. Main Environmental Factors Influencing Soil Microbial Communities in Alpine Wetlands

A redundancy analysis (RDA) was performed to determine the strength of the association between the soil microbial community and climate and soil physical and chemical properties (Figure 5). The first two canonical axes explained 21% (29% by RDA1 axis and 22% by RDA2 axis) and 58% (36% by RDA1 axis and 22% by RDA2 axis) of variance for bacterial and archaeal communities, respectively. The RDA indicated that TN, TP, TOC, SMC, and temperature had a significant influence on the bacteria community (*p* < 0.05); TN, TP, and TOC had significant effects on the archaea community (*p* < 0.05) (Table 3).

## 4. Discussion

Our results suggest that the composition of bacterial communities was strongly affected by seasonal changes. Significantly, the abundance of Proteobacteria at the surface depth (0–5 cm) decreased by 5.1%, 3.0%, and 1.3% in Winter compared with the other three seasons. Previous studies have shown that Proteobacteria prefer soil with a high nutrient content, which is significantly related to the content of soil organic matter and total nitrogen [28]. This study’s physical and chemical parameters suggest that the nutrient content at the 0–5 cm depth of soil was the highest, and Proteobacteria had the most significant seasonal changes in the 0–5 cm surface among the main bacterial lineages. In addition to the influence of soil nutrients in each soil layer, the actual temperature and precipitation data were measured by Yue et al. [29]. At the Namco station, the soil temperature is 2.94 (°C), 11.16 (°C), −1.01 (°C), and −9.89 (°C) in Spring (Spr), Summer (Sum), Fall (Fal), and Winter (Win), respectively. Precipitation is mainly concentrated in July-August (242 mm), and the soil temperature and precipitation are the “unimodal type”. Some studies have shown that the abundance of microbial communities is closely related to the seasonal changes in temperature and precipitation [30,31,32]. In this study, the relative abundance of Bacillota was 6.7% in Summer > 3.4% in Spring > 3.3% in Fall > 2.9% in Winter, and the relative abundance of Methanomicrobia was 4.2% in Summer > 3.2% in Spring > 2.7% in Fall > 2.2% in Winter. Because Bacillota are facultative anaerobes and Methanomicrobia are anaerobes, they are expected to maintain their growth more under anoxic conditions compared with other microorganisms. In Summer, 63% of the annual precipitation is concentrated in Namco [33], and the dissolved oxygen in the soil is low [34]; this may have inhibited the growth of other aerobic bacteria.

The higher bacterial diversity than archaeal diversity observed is similar to what is seen in other wetlands, such as the Huixian and Nanniwan wetlands in China [27,35]. However, some major microbial lineages may differ due to the different locations. Additionally, the bacterial Chao1 index suggests significant differences between Winter and the other three seasons, and the Chao1 index of archaea did not show significant differences with respect to seasons. At the same time, the soil moisture content (*p* = 0.001, <0.05) and temperature (*p* = 0.025, <0.05) also decreased significantly in Winter. A study on the Ruoergai wetland proved that soil moisture directly impacts microbial community and diversity [36]. Based on the results of studies on the effects of biome time series [37,38], temperature affects biological enzyme activity, and an extended frost period and freezing period also affect the nutritional status of soil and increase the environmental pressure on microorganisms. Therefore, the long-term low temperature in Winter might inhibit biological enzyme activity, thus affecting microbial metabolism and reducing the diversity of the microbial community. The changes in biological enzyme activity and soil microbial community structure in Tanggula Mountain show that biological enzyme activity is directly related to microbial community structure and diversity [39]. In this study, the relative abundance of *Thermoproteus* bacteria increased by 7.0% in Winter. It is speculated that some cryophilic microorganisms were more durable in Winter in the Namco wetland.

All the measured soil N and C cycling rates in Tibetan wetlands did not differ significantly among the wetland types. However, significant differences in the diversity and abundance of microbial communities were observed. Soil moisture and N and C availability could impact soil N and C cycling rates directly and indirectly by altering the microbial community structure.

The PERMANOVA suggests significant differences between the microbial community in surface soils and those in deep soils, which was elucidated by the microbial community structure with a layered distribution in the wetland soil of the Yellow River Delta [40]. In surface soil, the bacterial community structure is more complex. In subsurface soil, the bacterial community structure is relatively simple because there are fewer nutrients, and the survival conditions of microorganisms are harsher; archaea thus become the main component of the microbial community. In addition, the archaea community structure in subsurface soil is more straightforward than that in surface soil, which is mainly composed of methanogenic archaea and sulfur-reducing bacteria. This is because the oxygen content in deeper soils is lower, and archaea are usually anaerobic organisms that can survive and thrive in such conditions. Moreover, there are fewer nutrients in deep soil, and archaea are generally able to metabolize using simple organic and inorganic materials, so they are better able to survive and reproduce in these environments. These results suggest that the layered distribution of aerobic or anaerobic bacteria and archaea is noticeable in different soil depths, indicating that with an increase in soil depth, the difference in oxygen supply creates an aerobic or anaerobic environment, resulting in the regular layered distribution of the microbial community structure. Plant roots may also be one of the factors that influence the differences in microbial community structure at different depths [41].

RDA revealed that the soil microbial community structure is closely related to soil physical and chemical properties, and the impact of soil physical and chemical properties on bacterial communities is significantly higher than that on archaea. In particular, bacteria were significantly affected by soil moisture content (SMC) and temperature (T) in the Namco wetland. Many studies have proven that soil TN, TP, TOC, and soil water content are significant factors affecting the structure of microbial communities [42,43]. The seasonal and soil depth changes in the soil microbial community in this study area are mainly concentrated in the surface soil at a depth of 0–5 cm, except the nutrient content in the surface soil of the Namco wetland is high, and the water content is sufficient. Seasons may also have a significant impact on vegetation type [44]; the quality and quantity of C, N, and P [45]; and trace elements other than these elements [46]. However, the negative correlation between pH and bacterial and fungal abundance is inconsistent with previous studies, which reported that decreased pH inhibits microbial growth [47], which may be due to unique habitats such as low temperature and low nutrients in the alpine zone of the Namco wetland, causing the changes in special functional populations in this area to be different from those in other soil ecosystems. For instance, the dominance of Proteobacteria and Acidobacteria in acidic surface soils suggests that there are functional traits enabling survival under low-pH conditions. Specifically, Proteobacteria (e.g., Rhizobiales, Burkholderiales) likely drive nitrogen cycling through N_2_ fixation and denitrification, while certain lineages within this phylum contribute to organic matter decomposition via hydrolytic enzyme production. Acidobacteria exhibit oligotrophic lifestyles optimized for slow-growing, energy-efficient metabolism in nutrient-limited acidic soils, playing key roles in recalcitrant carbon decomposition. These taxonomic shifts have direct implications for biogeochemical cycling. The proliferation of acid-tolerant bacteria under decreasing pH may enhance organic matter mineralization rates, releasing labile C/N/P compounds that fuel primary productivity in surface soils. Conversely, the observed pH-related suppression of fungal abundance could reduce the competitive inhibition of bacterial decomposers, further altering nutrient turnover dynamics. For archaeal communities, although less responsive to environmental gradients than bacteria, methane-cycling lineages (e.g., *Methanomicrobia*) and ammonia-oxidizing Thaumarchaeota likely maintain critical roles in anaerobic carbon metabolism and nitrification, respectively [48].

## 5. Conclusions

In summary, based on the high-throughput sequencing of the 16S rRNA gene, this study depicted the seasonality and vertical patterns of soil microbial communities and their main drivers in the Namco wetland on the Tibetan Plateau. We found that the soil microbial community in alpine wetlands exhibited apparent seasonal variation, and the diversity and composition of the bacterial community in Winter were especially significantly different from those in the other three seasons, which the soil nutrients, temperature, and moisture may have caused. More interestingly, the differences in soil microorganisms under different soil layers are mainly reflected in Spring, when the community composition of bacteria and archaea in surface soil (0–5 cm) significantly differs from that in subsurface soils (5–30 cm). This may be mainly due to the lowest water level occurring in Spring and because the soil water content of the surface soil layer has not yet reached saturation. Overall, these findings highlighted the importance of seasonality variation and the vertical pattern of the microbial community in the alpine wetlands on the Tibetan Plateau.

## Figures and Tables

**Figure 1 microorganisms-13-00962-f001:**
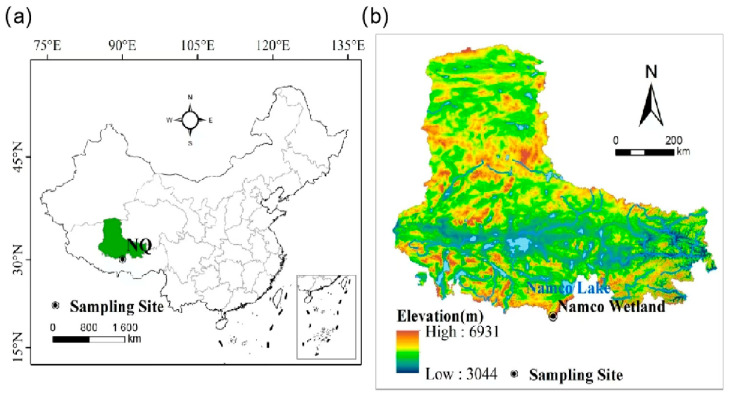
A map of the study area. (**a**) The study area is located in China, with the sampling site indicated by a black dot. (**b**) A topographic map of the study area shows Namco Lake and the Namco wetland.

**Figure 2 microorganisms-13-00962-f002:**
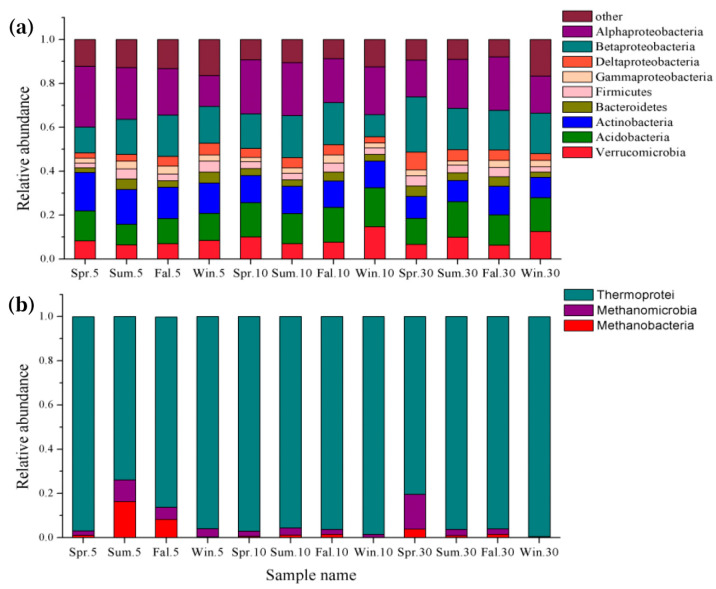
The relative abundance of the dominant phyla of the bacterial community (**a**) and the dominant classes of the archaeal community (**b**). Spr: Spring; Sum: Summer; Fal: Fall; Win: Winter. Soil depth abbreviations: 5: 5 cm, 10: 10 cm, and 30: 30 cm.

**Figure 3 microorganisms-13-00962-f003:**
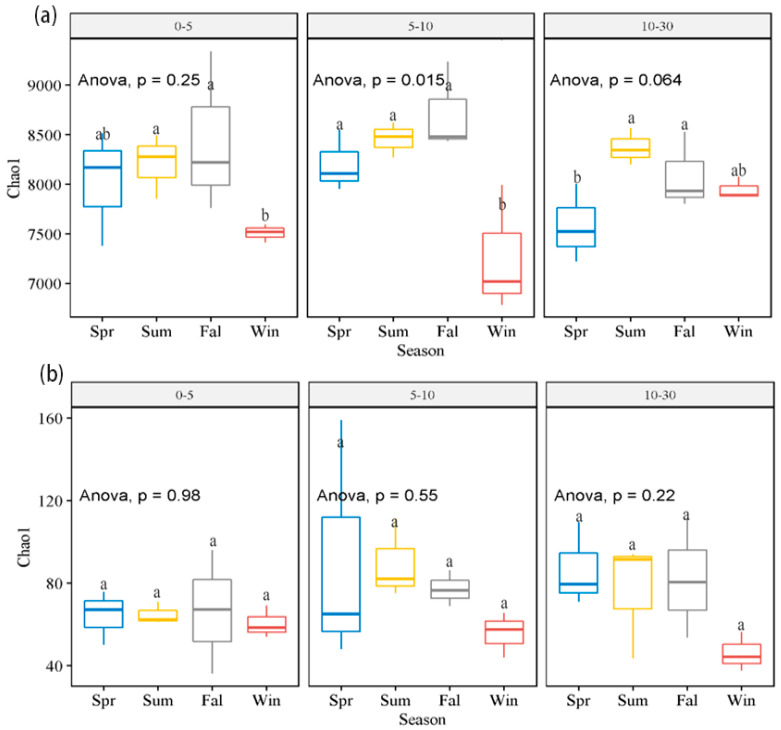
Microbial Chao1 index of different seasons at same soil depth of alpine wetlands. (**a**) Bacterial and (**b**) archaeal Chao1 index across different seasons and depths. Different lowercase letters indicate significant differences (*p* < 0.05) between seasons in same soil depth. Spr, Sum, Fal, and Win represent Spring, Summer, Fall, and Winter, respectively.

**Figure 4 microorganisms-13-00962-f004:**
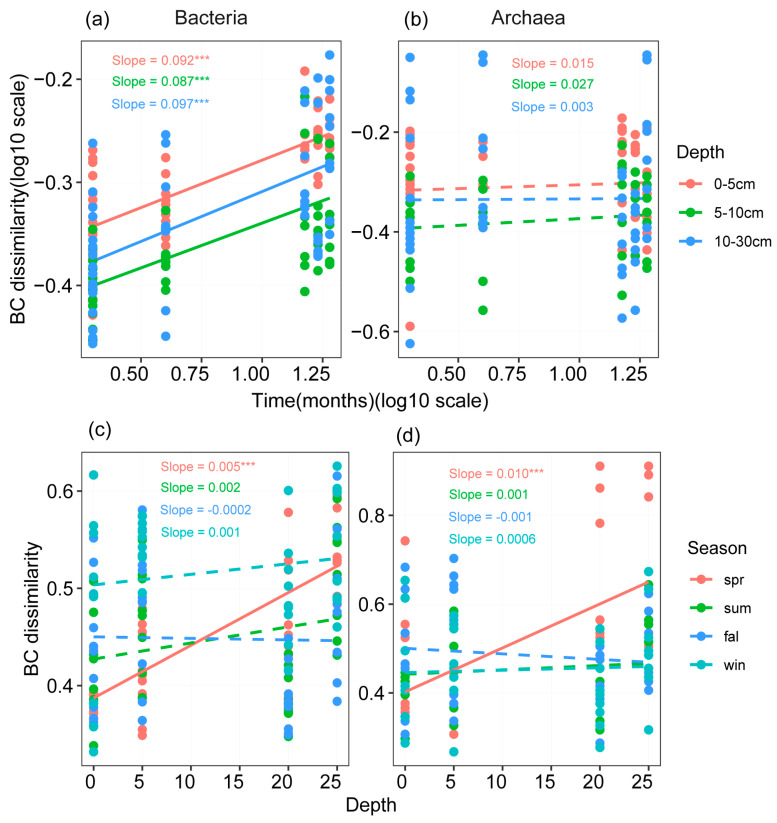
The dissimilarity of bacterial and archaeal communities across seasons and soil depths. The community dissimilarity was calculated based on the Bray–Curtis distance matrix. The solid and dotted lines represent significant (*p* < 0.05) and nonsignificant (*p* > 0.05) results, respectively. The slope represents a linear model comparing community dissimilarity and season distance (**a**,**b**) or soil depth (**c**,**d**). *** represent significant differences at the 0.001 levels, respectively. Spr, Sum, Fal, and Win represent Spring, Summer, Fall, and Winter, respectively.

**Figure 5 microorganisms-13-00962-f005:**
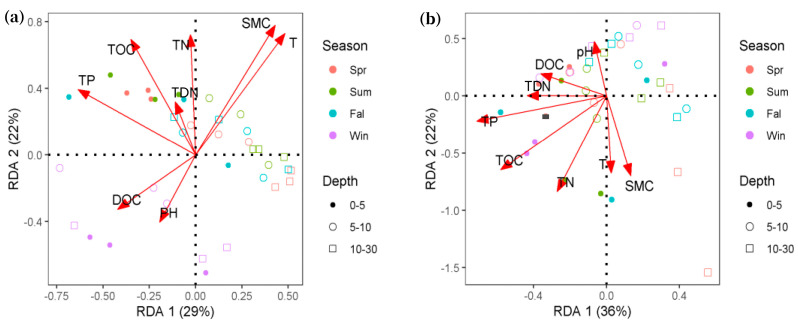
The redundancy analysis was conducted to explore the influence of environmental factors on soil bacterial (**a**) and archaeal (**b**) community structures in alpine wetlands. Spr, Sum, Fal, and Win represent Spring, Summer, Fall, and Winter, respectively. TN: total nitrogen; TP: total phosphorus; TOC: total organic carbon; DOC: dissolved organic carbon; TDN: total dissolved nitrogen; pH: soil pH; SMC: soil moisture content; T: atmospheric temperature.

**Table 1 microorganisms-13-00962-t001:** Soil physical and chemical properties across different seasons and depths in alpine wetlands.

Index Parameter	Sample Depth	Spring	Summer	Fall	Winter
TN (mg/kg)	0–5 cm	10.29 ± 1.91 ^Aab^	11.29 ± 1.10 ^Aa^	8.35 ± 0.43 ^Ac^	7.63 ± 1.30 ^Ad^
TP (mg/kg)		1.80 ± 0.22 ^Aa^	1.87 ± 0.20 ^Aa^	1.46 ± 0.19 ^Aa^	1.65 ± 0.21 ^Aa^
DOC (mg/kg)		26.88 ± 12.08 ^Abc^	18.34 ± 1.77 ^Ac^	37.15 ± 1.98 ^Abc^	41.38 ± 12.79 ^Aa^
TDN (mg/kg)		5.81 ± 3.05 ^Aa^	3.06 ± 0.31 ^Aab^	5.48 ± 1.16 ^Aa^	4.15 ± 0.46 ^Aa^
TOC (mg/kg)		0.15 ± 0.025 ^Aa^	0.14 ± 0.007 ^Aab^	0.10 ± 0.006 ^Ab^	0.09 ± 0.026 ^Ab^
pH		7.43 ± 0.01 ^Aa^	7.22 ± 0.16 ^Aa^	7.32 ± 0.07 ^Aa^	7.50 ± 0.25 ^Aa^
SMC (%)		91.47 ± 3.86 ^Aa^	121.77 ± 7.53 ^Aab^	139.71 ± 17.47 ^Aab^	88.39 ± 0.1.98 ^Ab^
TN (mg/kg)	5–10 cm	10.12 ± 0.85 ^Aab^	8.35 ± 2.80 ^Aab^	8.91 ± 0.31 ^Aab^	7.53 ± 2.66 ^bc^
TP (mg/kg)		1.66 ± 0.28 ^Aa^	1.44 ± 0.08 ^Aab^	1.45 ± 0.23 ^Aab^	1.69 ± 0.45 ^ABab^
DOC (mg/kg)		27.11 ± 6.71 ^Aab^	25.96 ± 4.78 ^Ab^	26.78 ± 10.23 ^Bb^	27.83 ± 12.09 ^ABab^
TDN (mg/kg)		4.51 ± 0.58 ^Aab^	5.40 ± 1.41 ^Aa^	3.76 ± 1.45 ^Ac^	4.54 ± 2.60 ^ab^
TOC (mg/kg)		0.12 ± 0.009 ^Aa^	0.09 ± 0.046 ^Aa^	0.09 ± 0.023 ^Aa^	0.72 ± 0.024 ^Aa^
pH		7.35 ± 0.09 ^Abc^	7.38 ± 0.13 ^Aabc^	7.28 ± 0.05 ^Abc^	7.60 ± 0.09 ^Aa^
SMC (%)		121 ± 22.72 ^Aa^	133.07 ± 7.37 ^Aa^	124.68 ± 22.97 ^Aa^	56.28 ± 20.47 ^ABb^
TN (mg/kg)	10–30 cm	8.03 ± 2.22 ^Ab^	8.74 ± 0.76 ^Aab^	8.96 ± 1.31 ^Aab^	4.93 ± 0.92 ^Bc^
TP (mg/kg)		1.34 ± 0.11 ^Abc^	1.46 ± 0.26 ^Aabc^	1.57 ± 0.24 ^Aabc^	1.31 ± 0.09 ^Ac^
DOC (mg/kg)		18.01 ± 1.51 ^Ac^	20.84 ± 4.98 ^Bc^	21.83 ± 2.51 ^Ab^	28.66 ± 6.64 ^Cab^
TDN (mg/kg)		3.37 ± 0.43 ^Aab^	3.45 ± 1.09 ^Aab^	3.22 ± 0.98 ^Aab^	1.02 ± 0.20 ^Bc^
TOC (mg/kg)		0.08 ± 0.041 ^Aab^	0.08 ± 0.009 ^Aab^	0.09 ± 0.037 ^Aa^	0.05 ± 0.019 ^d^
pH		7.42 ± 0.21 ^Aabc^	7.29 ± 0.02 ^Abc^	7.19 ± 0.25 ^Ac^	7.35 ± 0.09 ^Abc^
SMC (%)		111.12 ± 12.28 ^Ab^	136.41 ± 19.16 ^Aa^	111.45 ± 3.94 ^Ab^	34.15 ± 0.83 ^Cc^
T (°C)		2.94 ± 0.91 ^b^	11.16 ± 0.66 ^a^	−1 ± 2.68 ^b^	−9.88 ± 1.63 ^c^

Note: Spr, Sum, Fal, and Win represent Spring, Summer, Fall, and Winter, respectively. TN: total nitrogen; TP: total phosphorus; TOC: total organic carbon; DOC: dissolved organic carbon; TDN: total dissolved nitrogen; pH: soil pH; SMC: soil moisture content; T: atmospheric temperature. The data are represented as the mean ± standard deviation; different letters in the same columns (uppercase A, B and C) indicate significant differences among soil depths and in the same rows (lowercase a, b, c, and d) indicate significant differences among seasons (*n* = 3, *p* < 0.05).

**Table 2 microorganisms-13-00962-t002:** Soil microbial community structure revealed across different seasons and depths in alpine wetlands by using PERMANOVA.

Sample Name	Index Parameter		Mean Squares	F.Model	Variation	Pr (>F)
Bacteria	Soil depth (cm)	0–5/5–10	0.272	2.418	0.099	0.001 **
		0–5/10–30	0.348	2.898	0.116	0.001 **
		5–10/10–30	0.127	1.207	0.052	0.169
	Season	Spr/Sum	0.106	1.019	0.060	0.359
		Spr/Fal	0.155	1.480	0.085	0.067
		Spr/Win	0.295	2.438	0.132	0.002 **
		Sum/Fal	0.106	1.035	0.061	0.383
		Sum/Win	0.306	2.567	0.138	0.003 **
		Fal/Win	0.301	2.516	0.136	0.003 **
Archaea	Soil depth (cm)	0–5/5–10	0.254	2.860	0.115	0.007 **
		0–5/10–30	0.387	3.626	0.141	0.003 **
		5–10/10–30	0.172	2.054	0.085	0.061
	Season	Spr/Sum	0.064	0.598	0.036	0.755
		Spr/Fal	0.078	0.693	0.042	0.653
		Spr/Win	0.168	1.481	0.085	0.212
		Sum/Fal	0.051	0.541	0.033	0.882
		Sum/Win	0.154	1.635	0.093	0.114
		Fal/Win	0.102	1.030	0.060	0.361

Note: Spr, Sum, Fal, and Win represent Spring, Summer, Fall, and Winter, respectively. ** indicate significance level at *p* < 0.01.

**Table 3 microorganisms-13-00962-t003:** The influence of environmental factors on soil microbial community structure in alpine wetlands revealed by using redundancy analysis.

Environmental Variables	Bacteria			Archaea		
	Variance	F	Pr (>F)	Variance	F	Pr (>F)
DOC	0.01	1.24	0.13	0.01	1.05	0.34
pH	0.01	1.20	0.15	0.01	0.82	0.61
SMC	0.02	2.25	0.001 ***	0.02	1.00	0.53
TDN	0.01	1.41	0.06	0.01	1.45	0.12
TOC	0.01	1.42	0.05 *	0.02	2.30	0.01 *
TN	0.01	1.84	0.01 **	0.01	27	0.02 *
TP	0.02	2.22	0.01 **	0.01	3.00	0.01 *
T	0.02	2.21	0.001 ***	0.02	1.04	0.35
T	0.02	2.21	0.001 ***	0.02	1.04	0.35
	Inertia	Proportion		Inertia	Proportion	
Constrained	0.10	0.30		0.07	3.00	
Unconstrained	0.21	0.70		0.18	0.70	

Note: TN: total nitrogen; TP: total phosphorus; TOC: total organic carbon; DOC: dissolved organic carbon; TDN: total dissolved nitrogen; pH: soil pH; SMC: soil moisture content; T: atmospheric temperature. The F-value is a critical statistic in RDA used to assess the degree of explanation or influence of independent variables (environmental factors) on the dependent variable (microbial community structure). A larger F-value indicates a more significant impact of environmental factors on the microbial community structure. The *p*-value (Pr (>F)) is the probability of observing the current F-value or a more extreme value under the null hypothesis that there is no significant relationship between the environmental factors and the microbial community structure. A smaller *p*-value indicates a greater likelihood of a significant relationship between the environmental factors and the microbial community structure. * indicates significance when *p* < 0.05, ** *p* < 0.01, and *** *p* < 0.001.

## Data Availability

The 16S rRNA gene sequences used for this paper are available in the NCBI (www.ncbi.nlm.nih.gov) under BioProject No. PRJNA1026561 accessed by 10 October 2023.

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
