# Peer review of "Seasonality and Vertical Structure of Microbial Communities in Alpine Wetlands"

_microorganisms, 2025, doi:10.3390/microorganisms13050962_

Round 1
Reviewer 1 Report
Comments and Suggestions for Authors
Review for „Seasonality and vertical structure of microbial communities in the alpine wetland” written by Wang et al.
Importance of the study: Alpine wetlands are unique ecosystems characterized by extreme environmental conditions like low temperatures, fluctuating water levels, and high UV radiation. These conditions strongly influence the microbial communities that play crucial roles in nutrient cycling, organic matter decomposition, and overall ecosystem functioning. Understanding the seasonal dynamics and vertical distribution of microbial communities in alpine wetlands is essential for assessing their ecological significance and responses to environmental changes.
The authors provide a comprehensive analysis about the seasonal and vertical structure of microbial communities. The research data are well-analyzed and well-presented. The authors stated that the microbial communities were influenced by seasonal changes. The amount of bacteria and composition of the bacterial community in winter differed significantly from those in the other three seasons, which may have been caused by the soil nutrients, temperature, and moisture. In addition, the diversity of soil microorganisms under various soil layers is most noticeable in the spring, when the bacterial and archaeal community composition in surface soil differs greatly from that in subterranean soils.
Specific comments:
Line 12: “material transformation” does not sound scientific or specific. Did you mean nutrient recycling or organic matter decomposition? Please rewrite the sentence.
Line 17: Please delete “Here, we collected soil samples” Please rewrite the sentence and explain the goal and the hypothesis of the study. After that, the authors can present that they collected soil samples, how many samples, from where, etc.
Please arrange the keywords in alphabetical order.
The Introduction section provides a good background for the study.
Line 71: Please use superscript for hm2.
Line 74: Please correct about 60-80%, or from 60% to 80%
Lines 78. Could you please provide a figure about the monthly average temperature from 2017? Thank you.
Reviewer 2 Report
Comments and Suggestions for Authors
Abstract.
Line 30. Please insert “may exert” before “on”
Introduction.
The introduction is overloaded with study cases from China. Please include information generated outside China. In the introduction, the authors maintain a descriptive approach instead of a causation description of processes. Please include a more detailed description of processes including relevant taxa and their demonstrated physiological and ecological role in modulating bacterial diversity.
Line 53. Please substitute “act” with “play a relevant role”. Please substitute “of” with “in”. Please replace “ecological” with “nutrient”
Line 38. Please substitute a study case with a wider review (Reference 2) analyzing information worldwide, not only cases from China.
Line 40. Same case. The authors must avoid generalizing ideas and arguments from geographically limited studies. A search for “seasonal dynamics microbial diversity soil depth” in Google Scholar yields 197K results, so the last argument is weak. Authors must revise the literature to deepen their argument on a hypothesis-guided basis.
Line 55. In this section, the authors revise some studies on bacterial diversity shifts, however, no functional evidence is presented, only that for methanogenic bacteria. Authors must generate a hypothesis-guided justification instead of trying to justify the work only because of the “absence” (?) or previous local evidence.
Lines 63-67. Please revise the objectives. They are merely descriptive and must be changed to hypothesis-testing objectives.
Materials and methods
Line 71. Please express area in hectares or square meters
Line 72. Please change “m” for “meters above sea level”. Please use the whole word “Figure”
Line 73. Please use accepted scientific names like “Carex tibetikobresia” instead of "Kobresia tibetica” synonym, please visit https://www.gbif.org/ to verify the accepted name. Please add “(Cyperaceae )” before “with”
Line 79. Please use the whole word “Figure”
Line 96. Please change “was” with “were”
Line 98. Please change “is” with “was”
Line 99. Please substitute “baked” with “incubated”
Line 102. Please change “oC” with “°C”
Line 104. Please change “carbon” with “nitrogen” before (DON)
Line 107. “TDN” is not explained before. Please include “Total dissolved nitrogen” and explain how it was determined.
Line 111. Please cite all materials, reactants, and instruments as “name (catalog number, manufacturer, city and country of manufacturer’s headquarters)”
Lines 127-135. Please cite all software and packages employed.
Results
Lines 137-139. Please remove this paragraph.
Line 152. Table 1. To ease readability and analysis, please present values grouping by parameter instead of grouping by soil depth. Switch between columns 1 and 2. Please superscript significance letter.
Line 158. Please clarify what “The data were mean ± standard, different letters in the same column meant significant difference” refers to. In the current description, it may be understood that values from different factors were compared by column. Please present significance letters on the two ways, by season, by soil depth, and by factor.
Line 162. Please add a link to the public accession to the raw sequence data in this section. Please state the exact number of samples analyzed, the total bp obtained, and the quality of readings. Please present the rarefaction curves by sample and total as supplementary material.
Line 168. Please use the entire word “Figure”
Line 172. Please update the taxon names of Firmicutes and Bacteroidetes.
Line 171. Please substitute “much” with “significantly”
Line 176. Please use the entire word “Figure”
Line 186. Figure 1. Please update the taxon names of Firmicutes and Bacteroidetes.
Line 187. Please insert “Season abbreviations:” after “Spr”
Line 188. Please insert “Soil depth abbreviations:” after “5”
Line 197. Please use the entire word “Figure”
Line 198. Please use the entire word “Figure”
Line 215. Please use the entire word “Figure”
Line 216. Please use the entire word “Figure”
Line 217. Please use the whole word “Figure”
Line 224. Table 2. Please insert the word “Soil” before “depth”
Line 230. Figure 4. Please, for consistency, use the same color code for the season used in figure 3. Please insert the word “Soil” before “depth”. Please use capital letters for abbreviations in the legend of panels C and D (Spr, Sum, Fal, and Win) as they appear in Figures 2 and 3.
Line 232. Please substitute “months” with “seasons”. Please insert the word “Soil” before “depth”.
Line 236. Please substitute “temporal” with “seasonal”. Please insert the word “soil” before “depth”.
Line 248. Table 3. Please justify values vertically in columns considering the decimal point [Column Pr(>F)].
Line 256. Figure 5. For consistency, please use the same color code for the season in Figure 3. Please insert the word “Soil” before “depth”. Please use filled markers for datapoints.
Discussion.
The authors are encouraged to enrich the discussion by contrasting their results with research outside China. The current discussion is highly “endogamic” (4 out of 22 references in this section), limiting the scientific interchange of ideas with colleagues from other regions. There are similar alpine wetlands in different continents where researchers deal with the same questions. A comprehensive discussion must consider their findings.
Line 268. Please change the word “preferred the” with “are more likely to be found in”. Please use “environments” in plural.
Line 275. Please use capital letters in seasons like “Spring”
Line 277. Please explain what "unimodal type" means.
Line 279. Please update the name of “Firmicutes clade”
Lines 281-282. Please use capital letters in seasons like “Spring”
Line 282. Please update the name of “Firmicutes clade”
Line 283. Please substitute “can” with “are expected to”
Line 284. Please remove “and their relative abundance is higher in summer”
Line 299. Please change “activity” with “metabolism” after “microbial”
Line 303. Please italicize “Thermoproteus”
Line 319. Please change “is” with “may be explained”
Line 323. Please change “showed” with “suggests”
Line 324. Please substitute “very obvious” with “noticeable”
Line 338. Please clarify what “the input of regulated falling objects” refers to.
Comments on the Quality of English LanguageI would recommend a Professional English style revision to improve readability
Reviewer 3 Report
Comments and Suggestions for Authors
In the manuscript titled "Seasonality and Vertical Structure of Microbial Communities in the Alpine Wetland" the authors investigated how seasonal variation and soil depth influence microbial community composition in the NamCo wetland on the Tibetan Plateau. The study analyzed soil samples collected across four seasons at different depths (0–5 cm, 5–10 cm, and 10–30 cm) and examined the microbial diversity of bacteria and archaea using high-throughput sequencing of the 16S rRNA gene. The current structure of the manuscript is somewhat unclear and requires reorganization to emphasize the key aspects of the study. For instance, while the authors provide a detailed account of the relative abundances of microbial communities at the phylum level, the manuscript lacks an in-depth discussion of genus-level composition, which is crucial for understanding microbial dynamics and ecological functions in alpine wetlands. Implementing these changes will improve the manuscript’s quality and enhance its potential for publication in this esteemed journal.
- Section 2.2. Sample collection requires more details regarding the number of samples collected (biological replicates).
- Lines 137-139: What is the purpose of this text?
- Table 1: The full names of the seasons should be included.
- Line 162: Specify the number of raw sequences and how many remained after quality processing.
- PERMANOVA terminology should be standardized, as it is inconsistently written throughout the document.
- Discussion Section: While the text dedicates significant space to describing seasonal fluctuations in phylum-level taxa, it should better integrate how these shifts correlate with environmental factors such as soil temperature, moisture, and nutrient content, which were identified as key drivers of microbial distribution in the redundancy analysis (RDA). In fact, this section should be revised to focus on genus-level analysis once the authors incorporate the identified genera into the manuscript.
- Additionally, given the nature of the ecosystem and the distinct soil horizons analyzed, the discussion would benefit from a clearer connection between microbial community composition and ecosystem functions, particularly regarding the roles of dominant bacterial and archaeal genera in biogeochemical cycling.
The English could be improved to more clearly express the research.
Reviewer 4 Report
Comments and Suggestions for Authors
The article is well-written, the experimental setup is scientifically sound, and the results are discussed in a proper manner.
Line 104: Correct with "dissolved organic nitrogen (DON)".
Lines 137-139: Delete these sentences.
Line 143: For clarity reasons, I suggest using either autumn or fall in the whole text (including Tables and Figures).
Table 1: I do not think that it is clear whether the statistical analysis refers to the comparison between seasons or between different depths. I think that using capital letters or different colours, you could also add the missing analysis.
Table 1: I suggest also adding the precipitation level. Moreover, separate in a clearer way the temperature (last row of the table) from the other measures of the layer 10-30 cm.
Chapter 3.2: I wonder if the information on the total abundance of bacteria and archaea in the different samples, therefore considering the depth and the seasonality, could give some additional insight into the study of this specific microbial community, aside from the relative abundance data that were presented in the Results.
Line 198: Correct the sentence.
Table 3: Describe what F and Pr(>F) stand for.
Table 3: I suggest placing the acronyms in the same order in which they are found in the Table, or in alphabetical order.
Line 276: You stated: “precipitation is mainly concentrated in July-August (242 mm)”. Are these data the average precipitation in the single month considered or in the entire period, considering July and August together?
Round 2
Reviewer 3 Report
Comments and Suggestions for Authors
Thank you for addressing the comments.
Reviewer 4 Report
Comments and Suggestions for Authors
I have no further comments.